# Nexus of Quality Use of Medicines, Pharmacists’ Activities, and the Emergency Department: A Narrative Review

**DOI:** 10.3390/pharmacy12060163

**Published:** 2024-11-01

**Authors:** Tesfay Mehari Atey, Gregory M. Peterson, Mohammed S. Salahudeen, Barbara C. Wimmer

**Affiliations:** 1School of Pharmacy and Pharmacology, College of Health and Medicine, University of Tasmania, Hobart, TAS 7005, Australia; 2Department of Clinical Pharmacy, School of Pharmacy, College of Health Sciences, Mekelle University, Mekelle 1871, Tigray, Ethiopia; 3Marinova Pty Ltd., Cambridge, TAS 7170, Australia

**Keywords:** emergency department, quality use of medicines, pharmacists, clinical pharmacy services, collaborative care, partnered pharmacist

## Abstract

Acute care provided in the hospital’s emergency department (ED) is a key component of the healthcare system that serves as an essential bridge between outpatient and inpatient care. However, due to the emergency-driven nature of presenting problems and the urgency of care required, the ED is more prone to unintended medication regimen changes than other departments. Ensuring quality use of medicines (QUM), defined as “choosing suitable medicines and using them safely and effectively”, remains a challenge in the ED and hence requires special attention. The role of pharmacists in the ED has evolved considerably, transitioning from traditional inventory management to delivering comprehensive clinical pharmacy services, such as medication reconciliation and review. Emerging roles for ED pharmacists now include medication charting and prescribing and active participation in resuscitation efforts. Additionally, ED pharmacists are involved in research and educational initiatives. However, the ED setting is still facing heightened service demands in terms of the number of patients presenting to EDs and longer ED stays. Addressing these challenges necessitates innovation and reform in ED care to effectively manage the complex, rising demand for ED care and to meet government-imposed service quality indicators. An example is redesigning the medication use process, which could necessitate a shift in skill mix or an expansion of the roles of ED pharmacists, particularly in areas such as medication charting and prescribing. Collaborative efforts between pharmacists and physicians have demonstrated positive outcomes and should thus be adopted as the standard practice in improving the quality use of medicines in the ED.

## 1. Background

The quality and efficiency of patient care in the emergency department (ED) are of great interest to clinicians, governments, and the public in general. Areas of particular focus in the ED often include access blocks and increasing healthcare demand [1]. In addition, a potentially high prevalence of drug-related problems (DRPs) occurring in the ED setting is another concern [2,3]. By definition, any drug-related actual or potential event affecting a treatment goal is termed a DRP [4]. Overcrowding and busyness, complex medication use processes, and the acute and fast-paced nature of clinical care make the ED a risk-prone setting for the occurrence of DRPs [5,6,7]. Congestion in the ED may also be associated with delays in the timely administration of essential treatments and poorer health outcomes [8,9,10]. The ED is also a key area for the transition of care, and these transitions pose a significant risk for the occurrence of errors and DRPs [11].

Drug-related problems are one of the most frequently reported clinical incidents in hospitals. A previous review has reported that 2.5% of hospital admissions may be associated with medication errors [12], defined as “any preventable event that may cause or lead to inappropriate medication use or patient harm while the medication is in the control of the health care professional, patient or consumer” [13]. Two-thirds of medication histories may contain at least one medication error, with one-third posing a risk of harm [14]. Despite these significant burdens, about 70% of medication errors are deemed certainly or probably preventable [15], suggesting that early detection and resolution through the involvement of medication experts may be possible.

As medication experts, pharmacists have demonstrated their ability to improve patient and medication safety in Eds [16]. Studies have demonstrated that pharmacists are well skilled in obtaining the best possible medication history (BPMH) [17,18], which is a patient’s comprehensive list of medicines obtained using a patient or caregiver interview and confirmed with one or more reliable sources of medication information [19]. Early BPMH determination with the integration of pharmacists into ED care may improve medication safety [20,21]. Thus, improved patient care in the ED could be facilitated by the collaboration of pharmacists and doctors. 

This narrative review aims to present findings related to the nature and challenges of ED care, quality use of medicines (QUM) in the ED, and the roles and scope of pharmacists in the ED. The definitions, prevalence, causes, and burdens of medication errors in the ED are also presented.

## 2. Quality Use of Medicines in the Emergency Department

Quality use of medicines—which may alternatively be called medication safety—is defined as “selecting management options wisely, choosing suitable medicines if a medicine is considered necessary, and using medicines safely and effectively” [22]. 

There have been many initiatives to improve QUM across the globe. After the To Err Is Human: Building a Safer Health System publication in 2000 [23], medication safety has been a health priority in different countries. The High 5s Project was launched by the World Health Organization (WHO) in 2006 to address the continuing global issues of patient and medication safety by setting a 75% reduction goal in medication errors [24]. The name High 5s Project was derived from the program’s initial goal to decrease five safety problems for five years (2010–2015) in five countries. Other similar programs included The Netherlands’ Preventing Medication Errors in 2006 [25], the Institute for Healthcare Improvement’s How-to Guide: Prevent Adverse Drug Events in 2011 [26], the WHO’s Global Patient Safety Challenge: Medication Without Harm in 2017 [27] and the Pharmaceutical Society of Australia’s Medicine Safety: Take Care in 2019 [28]. In the context of preventing avoidable harms caused by medication errors, the WHO’s World Patient Safety Day 2022 theme was “Medication Safety”, with the slogan “Medication Without Harm” [29].

### 2.1. The Emergency Department

The care given in the ED is a key component of the healthcare system. The ED is the patient’s initial point of contact with acute healthcare providers and serves as the essential bridge between outpatient and inpatient care [30]. Despite its importance in healthcare, the ED is the riskiest setting for the occurrence of medication errors due to [31,32]: (a) the emergency-driven nature of the complex, undifferentiated disease, (b) the urgency of care required, (c) the presence of limited or no access to current and complete patient information on ED presentation, (d) rapid turnover of patients, (e) use of complex medication regimens, and (f) high rates of ED congestion.

### 2.2. Medication Errors in the Emergency Department

The existing literature indicates that medication errors are characterized in several ways [33]. Figure 1 shows definitions and relationships of the different terminologies. A DRP refers to any drug-related actual or potential event that affects the goal of therapy [4]. Adverse drug events (ADEs) are defined as “an injury resulting from medical intervention related to a drug” [34]. The term “DRP” is a broader concept that embraces different events, such as medication errors, adverse drug reactions (ADRs), and poor adherence. Medication errors are “preventable events leading to inappropriate medication use or patient harm” [13], whereas ADRs are “noxious and unintended responses occurring at normal doses of drugs” [35]. Incorrect or omitted drugs and incorrect dosage regimens are examples of medication errors. Any differences between hospital prescriptions and patients’ home medications are defined as medication discrepancies. [36] Medication discrepancies can be classified into three types [37]. The first category of discrepancies is “documented intentional discrepancies”, in which the addition, modification, or discontinuation of medicine is intentional and clearly recorded. The second category consists of undocumented intentional discrepancies, which involve the intentional addition, modification, or discontinuation of medicine; however, there is no clear documentation of the reasons in the patient’s medical record. Unintentional discrepancies, in which there are unintentional substitutions or omissions of a patient’s pre-admission medicine, constitute the third category of discrepancies.

A multicenter study conducted in nine hospitals reported that patients experienced different medication-related issues throughout their ED stay [2]. The prevalence of medication errors in the ED is about 23% [40]. Up to 50% and 35% of medication errors in the ED occur during the prescribing and administration stages, respectively [41,42]. A Cochrane systematic review (2018) that pooled results using 20 studies with 4629 participants showed that about half of patients at transitions of care may be exposed to medication errors [43].

The causes of medication errors in the ED are multifactorial. Overcrowding, increased workloads, or heterogeneity of medications may play a role in increased rates of medication errors in the ED [41,44,45]. Failure to accurately identify pre-admission medications may be another key contributor [46]. In a survey of patients, only 15% could correctly identify all of their medicines (including indications, frequencies, and doses) [47]. The majority of ED sources of individual patients’ drug information have inconsistencies and cannot be relied on in isolation [48].

Around half of the medication errors identified in the ED were found to be potentially harmful and might necessitate additional intervention or intensive monitoring [49]. Medication errors in the ED also resulted in 3% of all hospital-related adverse events [50]. Even though medication errors can be ubiquitously present in all hospital settings, preventable errors frequently occur in the ED [51], necessitating early detection and intervention before reaching the patient [2].

### 2.3. The Pharmacist in the Emergency Department

The ED pharmacy is a rapidly growing field of practice globally. Pharmacists’ involvement in the ED dates back to the 1970s when reports of ED pharmacy services began to emerge [52]. Different associations published guidelines/policy statements recognizing pharmacists’ participation in the ED; the American Society of Health-System Pharmacists in 2011 [53], the American College of Emergency Physicians in 2015 [54], the Society of Hospital Pharmacists of Australia in 2015 [55], the American College of Medical Toxicology in 2017 [56], and the Society of Infectious Diseases Pharmacists in 2017 [57].

Pharmacists’ roles in the ED have transitioned from basic inventory management to the provision of comprehensive clinical services. A review conducted by Cohen et al. 2009, summarizing the pharmacists’ activities described in the literature from 1976 to 2008, reported more than 50 tasks or responsibilities in the ED [16]. 

Generally, the scope of ED pharmacy services can be categorized into traditional roles, emerging roles, and others (Figure 2) [58]. Traditional roles are those services known to confer clinical benefits to ED patients based on previously well-established evidence. These roles are considered the practice standards of ED pharmacy services [55]. Emerging roles refer to advanced clinical activities of pharmacists with specialized training and responsibilities that are relatively new to the ED context [58]. Some examples of emerging roles include reviews of microbiological cultures, medication charting, and participation in resuscitation. Pharmacists in the ED are also involved in research, education, and ED-specific committees [58,59].

North American hospitals have been known for establishing and improving pharmacy services in the ED [60]. A study from a decade ago (Pedersen et al. 2013) showed that 16.4% of all USA hospitals had regularly assigned ED pharmacists, with 90% of them being involved in clinical services [61]. In Australia, pharmacy practice in the ED has been evolving since the 1990s [62]. A survey on ED pharmacy staffing was conducted in 57 Australian hospitals in 2016 [58]. Sixty-two per cent (35 of 57) of the hospitals had ED pharmacy services, with one-third having dedicated seven-day services, compared to 33 hospitals in 2010 [63] and 22 hospitals in 2003 [62], when the majority of services were offered part-time. Dedicated ED pharmacy services provided drug information, counselling, and clinical consultations to ED patients. In some hospitals, pharmacists working in the ED were also involved in resuscitation services, toxicology, and therapeutic drug monitoring. Some hospitals with dedicated ED pharmacy services provided charting of pre-admission medications (23.5% of hospitals) and new medications (35.4% of hospitals) [58].

Having a pharmacist in the ED may yield several layers of protection for a patient. There has been growing evidence of positive outcomes linked with pharmacists practicing in the ED by reducing medication errors, improving medication appropriateness and saving costs [20,64,65]. Clinical interventions conducted by ED pharmacists were related to considerable cost avoidance at the Intermountain Medical Center, Utah [66]. The ED pharmacists made 820 clinical interventions across 107 overnight shifts, leading to a substantial cost avoidance of USD 612,974, primarily due to direct patient care, ADE prevention, and individualization of patient care. Most ED staff have also positively accepted the pharmacy service in the ED with a high level of uptake of pharmacists’ recommendations [67].

Growing demand for ED pharmacy services and extension of the practice to emerging roles have been observed [58,68]. Despite these demands, several barriers and challenges associated with rendering the service in the ED still exist [69]. One challenge is the inaccessibility of ED pharmacy services, which are mainly concentrated in metropolitan hospitals and hospitals with a large number of inpatient beds [58]. In smaller hospitals, the availability of services such as round-the-clock pharmacist coverage and comprehensive MedRec is often limited due to resources and staffing constraints. These limitations can lead to delays in care, an increased risk of ADEs, and less comprehensive medication management. Regulatory differences, lack of prioritization of the services, resource scarcity and funding constraints also remain key barriers to the broader implementation of ED pharmacy services. There has been a continuous debate on how to implement ED pharmacy services that would improve QUM and the need to allocate resources for the service [53]. 

Staffing and resource constraints present significant barriers to the broader implementation of ED pharmacy services, particularly in resource-limited settings. The services could be optimized by reallocating specific tasks, for example, distributing nonclinical tasks to pharmacy technicians and higher level clinical interventions to pharmacists. Tele-pharmacy services could also be leveraged to extend the reach of pharmacists into EDs in resource-limited areas.

## 3. Interventions Improving Quality Use of Medicines in the Emergency Department

There are various types of ED-based interventions aimed at improving QUM in the literature. A broad classification can be made between pharmacists’ interventions and nonpharmacist-based interventions. Depending on the degree of involvement in ED care, the breadth of pharmacists’ interventions varies widely [20]. The following sections present key features, advantages and limitations of each intervention, their impact on QUM, and gaps in the literature.

### 3.1. Standardizing Medication Use Process

The medication use process is complex and involves about 20 steps, with 20 possible opportunities for an error to occur [70]. This multistep process can be categorized into subprocesses of prescribing, transcribing and documenting, dispensing, administering, and monitoring stages [71]. Standardizing each medication use process, such as unit-dose dispensing, has been recommended as one strategy for reducing medication errors. For instance, a previous study found that unit-dose packaging was reported to be effective in reducing medication errors from 24.3% to 9.7% compared to a stock-distribution system [72].

Standardizing intravenous infusions using smart pumps [73], standardizing labelling of similar drugs using “tall man” letters (e.g., hydrOXYzine, hydrALAZINE, DOPamine, DOBUTamine) [74] and limiting abbreviations and acronyms [75] are other strategies used to minimize medication errors. Although these measures have been known to safeguard patients’ safety, medication errors still pose a significant threat to ED care. 

### 3.2. Automated Dispensing Systems

Automated dispensing systems are medication-storage systems that facilitate the dispensing and storage of drugs at the point of care by a computer [76]. Even though the automated dispensing systems were hypothesized to improve medication efficiency and safety [77], the current evidence showed contradictory results: they were less effective in preventing errors, or they caused new types of errors, including medication selection errors. Additionally, clinicians may bypass safety features or override alerts, causing override errors [78,79]. Some important safety measures may also be overlooked if there is excessive reliance on automated technologies. It is, thus, imperative for institutions to constantly review their automated systems and verify that the risks are kept to the minimum while maximizing the benefits. 

### 3.3. Barcoding Medication Administration

The barcode medication administration system, initially implemented in the USA in the 1990s [80], is a system that labels an exclusive barcode identifier on each drug. This system improves the safety and quality of the medication administration process in the ED [81,82]. Scanning of the code allows the correct drug administration, thereby eliminating medication errors before reaching the patient. 

While the system exhibits some advantages, its potential to function as a medication safety strategy is limited by its shortcomings [83]. The generation of new types of errors is more likely to occur when labelling is different in primary packages and individual units. This mislabeling was reported as the most common source of barcode-related mistakes, causing 27% of medication errors [84]. Other safety safeguards may be overlooked when there is an excessive reliance on technology, which may sometimes galvanize the perpetuation of medication errors across the continuum of care. As with any technology, technical and human issues should be considered during implementation. The barcoding approach should be viewed as a process requiring continuous training of clinicians and amendment of the system. Using electronic systems can open new challenges in ensuring QUM in a complex and dynamic healthcare system [85,86].

### 3.4. Computerized Provider Order Entry Systems

A computerized provider order entry (CPOE) system is a type of patient management software that allows practitioners to enter medical instructions into a computer system [87]. An advanced CPOE system may include recommendations and information on drug interactions, dosing in renal insufficiency, doses for the elderly, and clinical laboratory testing. This may eliminate medical and medication errors that can arise from conventional verbal and handwritten orders. Evidence from CPOE studies has demonstrated a reduction in medication errors [88,89]. 

While CPOE is endowed with advantages, this system is not risk-free and is associated with numerous error types [90,91,92]. New errors include ignored antibiotic renewal notices and inflexible ordering formats, which can generate incorrect orders [90,93]. A false sense of security caused by automation contributed to an increased death rate following the implementation of a new CPOE system in a hospital providing tertiary care [94]. Alert fatigue and workflow interruption due to repeated warnings and alerts were also reported, causing these messages to be overlooked or superseded. Integrating these systems into a complex and dynamic medical environment requires vigilance. The system is only as good as its programming, interface, and design, as with any technical intervention. Without reasonable tutoring and training, it might not be an effective intervention type in reducing medication errors. The CPOE system may also require a concatenated validation process. It is important to note that the safety features ensured by these systems may be eluded in a busy ED setting, where most of the instructions are orally communicated.

### 3.5. Educational Initiatives

Insufficiency of drug knowledge and the addition of many new medications into the healthcare market necessitate the development of ongoing educational tools targeting staff development and work structures. A three-month, pre–post interventional study employing educational interventions on ED staff showed a selective improvement in certain medication error types [95]. Since most medication errors originate in the prescribing stage [41,42], educational interventions given to prescribers may have more impact in acute settings, such as the ED. For instance, a study employing education interventions directed at ED resident physicians found a statistically significant reduction in the rate of ADEs and the number of dose adjustments [96]. Another prospective pre–post study that used educational tools as an intervention to enhance ED nurses’ drug knowledge and awareness reported a statistically significant reduction in medication error rates from 34.2% to 15.3% [97]. ED pharmacists also play a significant role in ED staff training and education [98]. Still, little evidence exists linking these educational interventions to changes in patient and medication safety in the ED.

### 3.6. Medication Reconciliation

Medication reconciliation (MedRec) is a process of obtaining the most complete, up-to-date, and accurate list of medicines on every episode of patient care, with a formal process of reconciling and documenting medication changes, including reasons for these changes [99]. It should be documented by clinicians at least on hospital admission, transfer, and discharge [100]. Many healthcare organizations have championed MedRec as an important medication safety strategy. The Joint Commission on Accreditation of Healthcare Organizations adopted it as a national patient safety goal in 2005 [101]. Following this, the WHO and its collaborating agencies endorsed MedRec as a medication safety strategy [99,102]. Different types of MedRec strategies (e.g., electronic tools) have been used to ensure the patient’s safety during transitions of care [103,104].

Figure 3 illustrates the four steps for completing the MedRec process on every episode of care [105]. Obtaining the BPMH is the foremost step, followed by confirmation of its accuracy, reconciliation of the history with the prescribed medications, and continuous transmission of verified information. The BPMH is not a routine preliminary medication history but rather a systematic process of obtaining and verifying all the patient’s medications. Sources of medication information include patients’ medication lists, patients’ own medications, dose administration aids, community pharmacy dispensing histories, general practitioner letters, and/or residential care facility charts [19]. 

Although MedRec has been recognized as the prime strategy for detecting medication errors during the transitions of care [106,107], the errors’ clinical significance and MedRec’s benefits on workload reduction are poorly understood [108,109]. Different systematic reviews reported the impact of MedRec in the ED and during transitions of care [109,110,111]. Patient-related outcomes (except for improving posthospital healthcare utilization) and workload improvements have not been consistently demonstrated in the literature [110]. There is a lack of substantial evidence in the literature as to which MedRec strategies reduce medication errors most effectively [112]. Also, MedRec was found to be less effective at multiple transitions of care compared to a single transition of care (i.e., at either admission or discharge) [111]. Reported barriers to the effective implementation of a MedRec service were rapid turnover of patients, inadequate access to records, and patients’ communication and cognitive barriers [113,114,115]. These barriers have been known to limit MedRec’s effectiveness, which warrants the introduction of novel approaches and their investigation. The existing literature lacks consensus on best MedRec practices, and variability in MedRec implementation can impact its effectiveness. Future research should focus on developing and evaluating standardized MedRec protocols specifically tailored to ED settings.

### 3.7. Prescribing Roles

The roles of pharmacists in medication prescribing (also known as medication charting or medication ordering) have been growing. Safety checks by deploying pharmacists in the ED are important measures to guarantee that the right patient is taking the right medication at the right time [70]. The inclusion of pharmacists in the prescribing models has gained popularity in Australia, Canada, the UK, and the USA [116,117,118,119,120]. A few studies have investigated the physician–pharmacist prescribing models in the ED [3], surgery department [121], and sexual health clinics [122].

### 3.8. Sticker Model

A pre–post-intervention study was trialed in three hospitals’ ED to investigate a sticker model’s effectiveness [123]. This model, wherein pharmacists place stickers on medication charts to highlight potential issues or suggest therapeutic adjustments, was implemented to enhance communication between pharmacists and physicians. ED pharmacists charted medications and wrote their therapeutic suggestions on a removable sticker. The sticker contained therapeutic recommendations to doctors and information to nursing staff precluding drug administration before the issue on the sticker was resolved. The sticker was placed to cover the spaces in the signature sections. Prior to signing the medication chart, the doctors reviewed the recommendations before removing the sticker and making any changes. The study also demonstrated an improvement in the timeliness and accuracy of medication charts.

### 3.9. Supplementary Prescribing Model

A study was conducted in the ED of a referral hospital that compared a supplementary pharmacist prescriber model and a physician prescribing model [3]. In this model, an ED pharmacist acted as an “opportunistic prescriber” and charted pre-admission medications. With this intervention, the rate of error-free medication charts was 90% in the supplementary pharmacist prescriber model compared to 26% in the physician prescribing model [3]. Moreover, there was a lower medium-to-very-high risk of medication errors in the intervention group than in the control group. However, the study’s external validity is questionable because of the small sample size (i.e., 17 patients).

### 3.10. Co-Prescribing Model

Previous studies emphasized the increasing impact of pharmacists when involved earlier in the patient’s hospital admission and coprescribing process [120,124,125,126]. An example is the partnered pharmacist medication cocharting (PPMC) model of care, in which the pharmacist has an extended role going beyond traditional roles compared to usual care [120]. As an initiative to close the loop between the process of medication charting and MedRec, cocharting has gained recognition globally [124,127]. The term “partnered pharmacist medication charting”, an initiative developed by Alfred Health (Victoria, Australia), was coined in November 2012 as part of redesigning total quality care [120]. It was then replicated and expanded into other hospitals [125,126]. Other models demonstrating similarity in key features and effectiveness to PPMC are the collaborative medication charting model and pharmacist medication charting model [21,124].

Activities in the PPMC arm and the usual care arm (i.e., ED physician model) for one study [128], are shown in Figure 4. Credentialled pharmacists took BPMH early in the ED presentation, undertook timely MedRec, held direct clinical discussions with ED physicians and nursing staff, developed a collaborative medication management plan, and charted medications using the plan [120]. In partnership with the physician, the PPMC pharmacists also assessed each patient prior to admission for the need for venous thromboembolism (VTE) prophylaxis. The key feature of the PPMC was the involvement of pharmacists in medication cocharting. The teamwork allowed for better incorporation of the knowledge and skills of pharmacy professionals into patient care.

Pharmacists need additional training and certification to assume expanded roles in medication charting and coprescribing in EDs. Typically, this may involve advanced clinical training, such as a medication charting credentialling program [128]. Moreover, collaborative practice agreements or coprescribing privileges may necessitate certification in drug therapy management or the completion of further training in patient assessment and pharmacotherapy.

The impacts of ED-based PPMC compared to standard traditional medical charting have been reported in our previous studies conducted in a real-world ED setting.

A lower proportion of patients in the PPMC group exhibited at least one error (3%), in contrast to the usual care group (61%), and the number of patients needed to be treated with PPMC to prevent at least one high/extreme error was 4 (*p* = 0.03) [128].Use of at least one potentially inappropriate medication upon ED departure was significantly lower for the PPMC group compared to the usual care group (*p* = 0.04) [129].The median time from ED presentation to the time of critical medicines’ first dose administration in the PPMC group was 8.8 h, compared to 15.1 h in the usual care group (*p* < 0.001) [130].PPMC was associated with a higher proportion of patients having complete medication orders and receiving VTE risk assessments in the ED (both *p* < 0.001) [130].The median relative stay index, a measure of total hospital stay that considers patient complexity, was decreased by 15% with PPMC compared to usual care [131].The cost of PPMC care per patient to avert at least one high/extreme-risk error was approximately AUD 138. PPMC also led to an average cost saving of approximately AUD 1269 per admission per patient [131]. Cost savings estimations often include calculating the cost avoidance associated with preventing medication errors or reducing adverse drug events. Cost savings were generally based on the estimated costs of managing preventable errors or adverse outcomes, such as prolonged hospital stays [131]. These savings can vary significantly across different hospital settings, depending on factors such as patient demographics, hospital size, available resources, or the specific scope of pharmacist involvement.

As demonstrated in different efficacy, safety. and feasibility studies in ED and non-ED settings, the PPMC intervention is effective, safe, and achievable [120,125,126,130,131,132]. Evidence demonstrates not only the benefit of PPMC for medication safety but also stakeholder support of the model. Clinicians valued the PPMC model, supporting it with positive feedback [133]. Improvements in relationships between physicians and pharmacists were reported [120]. In summary, the PPMC has been shown to be a promising model that improves the accuracy of medication regimens on admission, optimizes patient care, and fosters multidisciplinary collaboration.

## 4. Discussion

Medication errors are more likely to occur in acute care settings with heavy workloads, such as the hospital ED, where the management and timing of patient care are critical and urgent [65,134,135,136,137]. Medication errors are prevalent in the ED; yet these errors are recognized or addressed only 20–-50% of the time [138]. The errors may lead to avoidable patient harm unless rectified or promptly identified [139]. The degree of patient harm could range from insignificant to compromised quality of life and increased risk of hospital admission from ED, mortality or greater health expenditures [139,140,141,142].

Even though early identification and resolution of medication errors in the ED before causing any harm to the patient is imperative, there are challenges in providing optimal ED care. Traditionally, physicians and nurses have been responsible for medication history taking in the ED. These professionals are already engaged with multiple pre-existing duties, i.e., routine medical and nursing care, and thus, medication-related issues may be overlooked that, in turn, increase the risk of errors and patient harm [2,143,144,145]. 

Although pharmacists are highly suited to obtain complete BPMH and review medications [17,146,147], restricted availability of pharmacy services in the ED is the second challenge [58]. According to a previous study [58], patients’ medications are not usually reviewed in hospital EDs without dedicated ED pharmacy services. This may create a significant delay in obtaining the BPMH. ED physicians may not even have access to the patients’ BPMH before writing the inpatient medication chart.

Another challenge to optimal ED care is related to the difficulty of obtaining a BPMH in a timely and complete manner at the point of initial encounter in the ED because of the patients’ disease acuity and the difficulty in interviewing them [148,149]. The activities are often prioritized to a few patients based on the availability of pharmacy staffing [58]. Without these early ED activities, medication-related issues may still be perpetuated throughout the inpatient stay and beyond [150]. 

Delays in implementing pharmacist recommendations have also been frequently cited as a barrier to optimal ED care [58]. Medication-related issues are often addressed retrospectively, sometimes after the issue has reached the patient. Pharmacists’ recommendations for enhancing QUM are sometimes not actioned, and patients can miss essential medications. 

### 4.1. Redesigning ED Patient Care

The ED setting is facing an increasing service demand [151]. Two recent time-motion studies of ED physicians in Norway (2022 and 2024) that quantified ED physicians’ activities reported that only 18% of physicians’ time was spent on drug-related activities, with 8 to 9 min per patient spent on medication history taking and documentation [152,153]. The frequency of interruptions experienced by ED physicians during drug-related tasks was higher compared to the interruption rate during nondrug-related tasks [152]. This is important because when ED physicians were required to multitask or were interrupted while prescribing, a two- to threefold increase in medication errors was observed [44]. Obtaining and documenting BPMH is also a complex, time-consuming process that requires a systematic approach. Therefore, when dedicated pharmacists undertake this task, the workload of busy physicians in an overcrowded ED setting may be lessened.

The evidence presented so far can add valuable information about redesigning and optimizing the flow of work and information in a fast-paced ED environment. Innovation and reform in ED care are driven by the need to manage the complex, rising demand for ED care and to meet government-imposed service quality indicators [154]. Redesigning the ED medication-use process could necessitate a shift in skill mix or an expansion of the roles of ED pharmacists [155,156].

If available, ED pharmacy services are often prioritized to a few patients with multiple comorbidities or complex regimens, based on the availability of pharmacy staffing [58,157]. Yet, the necessity of devoting resources to ED pharmacy services and the optimal method of integrating pharmacists into ED patient care has been the topic of ongoing debate [53]. In an era of rising demand for healthcare, the scope of pharmacy practice in the ED needs to be broadened [158]. Similar to our study, evidence on redesigning ED care by incorporating a pharmacist in medication coprescribing/cocharting model in the ED has demonstrated benefits in Australia and elsewhere [3,124,159,160,161]. Recent literature also shows that ED personnel have begun to recognize the safety benefits of incorporating pharmacists into the team, helping to reduce potentially harmful events and increase practice efficiency for patients seeking urgent care [158,162,163].

### 4.2. Closer Interprofessional Collaboration in ED

Collaborative efforts between pharmacists and physicians when working in tandem were most often associated with positive patient outcomes [164,165]. A case in point is the development of mutual trust, professional relationships, and the recognition of each other’s competencies and skills. Importantly, the team members shared a common focus—the patient. 

Involving pharmacists in (co-)prescribing roles can be viewed as evidence of encouraging teamwork and communication among interprofessional ED staff, particularly the pharmacy, medical, and nursing staff. The clinical conversation between ED pharmacists and ED doctors is important to codevelop a patient’s treatment plan, which can lead to better coordinated care for a patient. The discussion should center on a patient’s relevant clinical and medication information, allowing appropriate prescribing of medications, quicker administration of time-critical medicines, and timely completion of relevant investigations. It has been shown that verbal discussions between clinicians are more effective than written advice or retrospective prescription order assessments [123,133,156,166]. Perhaps direct verbal communication may facilitate acceptance of pharmacists’ recommendations when pharmacists are actively involved in interdisciplinary team-based care. Collaborative efforts between pharmacists and physicians, which have consistently been associated with positive patient outcomes, should be integrated as the standard of care in medication management within the ED.

### 4.3. Limitations and Future Work

Our review is a narrative review in nature. Our focus was primarily on the literature pertinent to our central themes of QUMs in ED settings and on studies examining the services provided by ED pharmacists. We aimed to highlight the most recent developments affecting medication safety and patient care in the ED setting. While narrative reviews provide summaries of existing literature and can be valuable in certain contexts, their lack of a reproducible search strategy renders them less rigorous and more prone to selection bias compared to systematic reviews. Future work should adopt a more structured method for data extraction and synthesis.

Our review focuses on interventions where pharmacists contribute significantly to critical judgment and patient care outcomes. Although pharmacy technicians play an important role in medication preparation and distribution, optimal patient care often arises from team-based approaches. Some of the pharmacists’ activities highlighted in our review could indeed be performed as part of an interdisciplinary team [20]. Future reviews should investigate how the involvement of pharmacy technicians enhances ED pharmacy services, particularly regarding operational efficiency and support in ED settings.

While the core principles of ED-based pharmacy services, such as improving medication safety and reducing errors, are broadly applicable across countries and settings, we acknowledge that the implementation and effectiveness of these services may vary across different healthcare settings. Factors such as staffing levels, access to technology, regulatory frameworks, and interprofessional collaboration models can influence how feasible and impactful these services are in other countries. Future research and localized studies would be crucial for ensuring the effective adaptation of pharmacist interventions in other ED settings.

Most studies emphasize short-term outcomes, such as the immediate reduction of medication errors. However, there is a need for data from longitudinal studies and follow-ups that assess the long-term effects on patient health, hospital readmission rates, and overall healthcare utilization. Future investigations should aim to provide more robust evidence on how ED pharmacist services contribute to improving patient outcomes over extended periods.

## Figures and Tables

**Figure 1 pharmacy-12-00163-f001:**
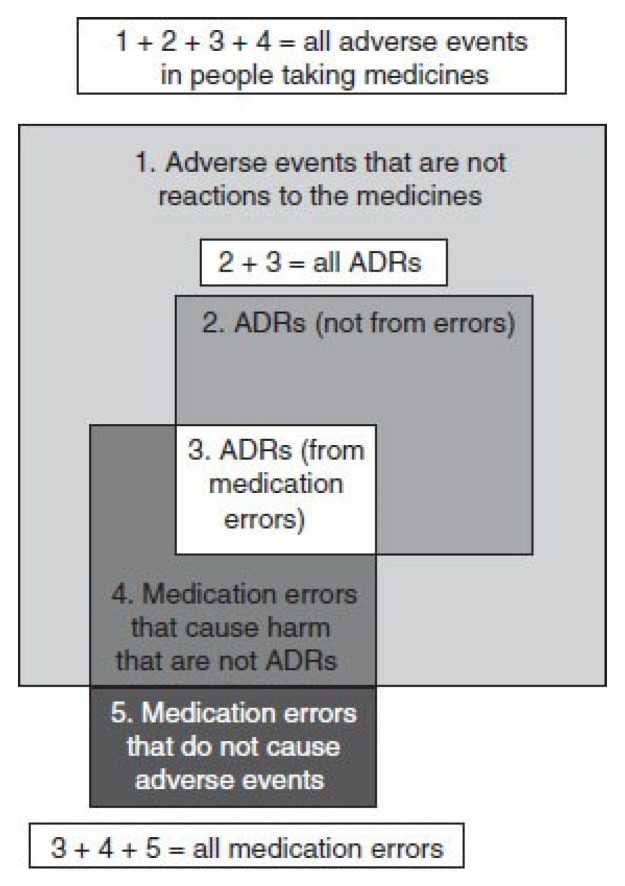
Relationships between ADRs, adverse events, and medication errors. Abbreviation: ADRs, adverse drug reactions. Sources: Aronson and Ferner, 2005; Ferner and Aronson, 2006 [38,39]; reprinted with permission from Springer Nature (License Number: 5451831388619).

**Figure 2 pharmacy-12-00163-f002:**
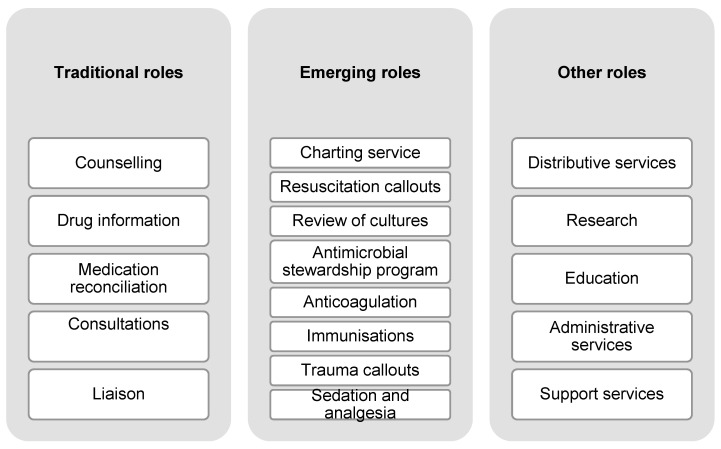
Summary of pharmacists’ roles in the emergency department. Source: Roman, et al., 2019 [58]; reprinted with permission from John Wiley and Sons (License Number: 5479200516913).

**Figure 3 pharmacy-12-00163-f003:**
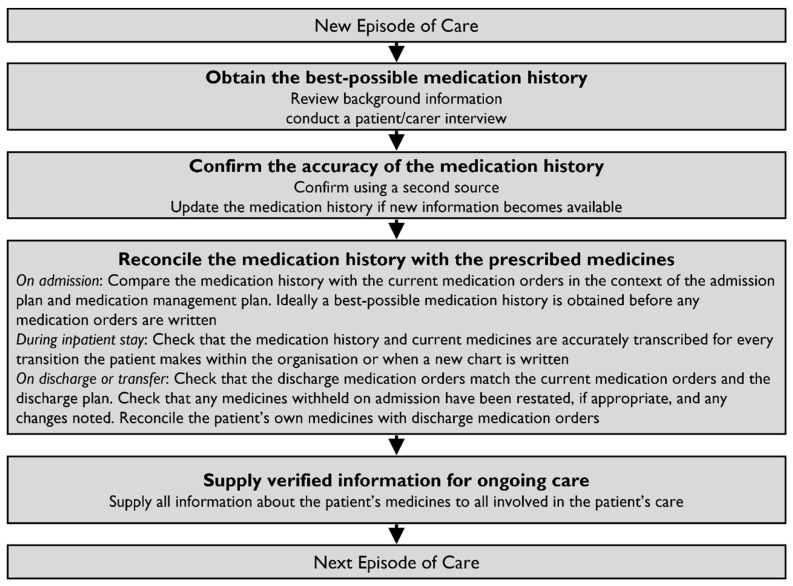
Medication review and medication reconciliation pathway. Source: The Society of Hospital Pharmacists of Australia [105]; reprinted with permission from John Wiley and Sons (License Number: 5479210066664).

**Figure 4 pharmacy-12-00163-f004:**
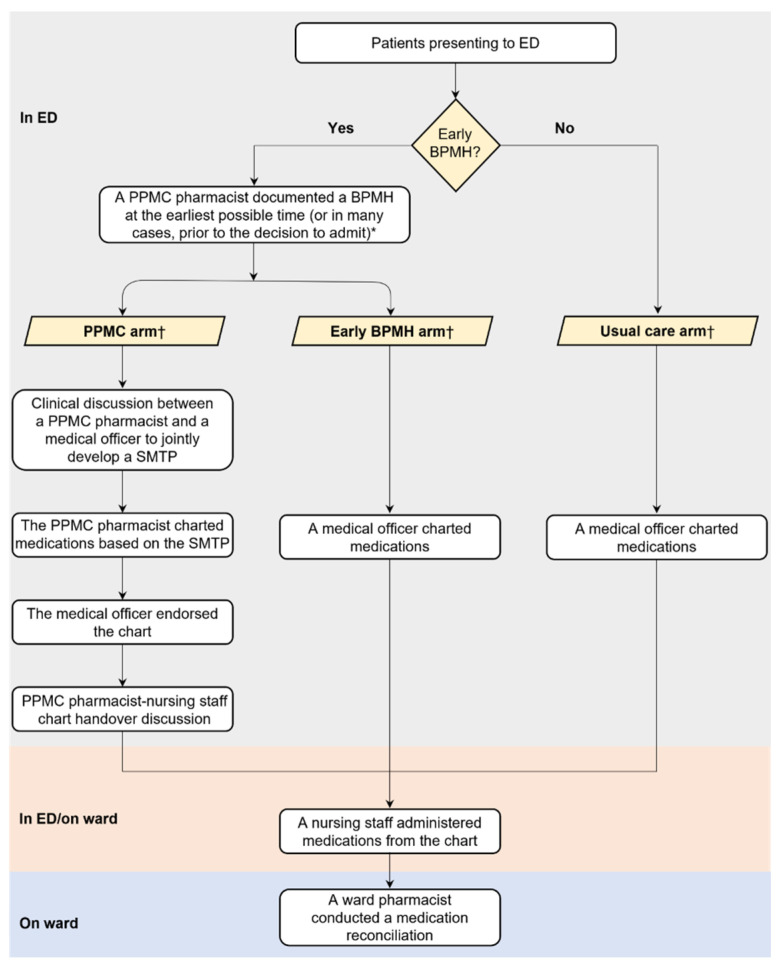
Traditional versus partnered pharmacist medication charting models. Source: Atey et al., 2023 [128]; reprinted with permission from the authors. Abbreviations: BPMH, best-possible medication history; ED, emergency department; PPMC; partnered pharmacist medication charting; SMTP: Shared medication treatment plan.

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
