# Peer review of "Nexus of Quality Use of Medicines, Pharmacists’ Activities, and the Emergency Department: A Narrative Review"

_pharmacy, 2024, doi:10.3390/pharmacy12060163_

Round 1

Reviewer 1 Report

Comments and Suggestions for Authors

The manuscript aims to review prevalence and efficacy of selected services performed by pharmacists or in collaboration of pharmacists and physicians aimed at safeguarding safe use of medication within emergency departments. The authors present results of previous studies showing the influence of various services on the quality use of medications. The manuscript focuses on the most recent developments in the area that could help to reduce the incidence of medication errors and improve provision of patient care in ED settings.

The main strength of the manuscript are as follows: a comprehensive review of various services and their impact on medication safety, extensive review of newest articles concerning the topic as well as thorough presentation of the positive impact of these interventions for both patients. Data presentation is clear and consistent, the article is well written. The cited literature is well chosen and mostly up to date. The conclusions are consistent with the presented evidence, the discussion refers to all important data reviewed in previous sections and is easy to follow.

I would raise some points that could be better addressed by authors:

1)     In lines 173-178 authors mentioned beneficial effects of pharmacists in ED regarding the cost reduction and increase of medicine appropriateness. Authors could refer to some specific findings in the cited literature as to provide some qualitative data concerning these improvements?

2)     In lines 182-184 authors point out that some ED services are only available in larger, centrally located hospitals, as compared to smaller hospitals. Which services are less accessible and how it influences the quality of medicines use?  

3)     In lines 214-215 authors wrote that previous studies indicated that the use of automated dispensing systems may lead to occurrence of new types of medication errors? What kind of errors become more common upon introduction of these systems?

4)     In lines 318-325 authors described the Sticker model and its introduction. I think it could be interesting to the readers of Pharmacy to know how it worked out. What were the main findings of the cited study?

Author Response

  1. Comment: The manuscript aims to review prevalence and efficacy of selected services performed by pharmacists or in collaboration of pharmacists and physicians aimed at safeguarding safe use of medication within emergency departments. The authors present results of previous studies showing the influence of various services on the quality use of medications. The manuscript focuses on the most recent developments in the area that could help to reduce the incidence of medication errors and improve provision of patient care in ED settings. The main strength of the manuscript are as follows: a comprehensive review of various services and their impact on medication safety, extensive review of newest articles concerning the topic as well as thorough presentation of the positive impact of these interventions for both patients. Data presentation is clear and consistent, the article is well written. The cited literature is well chosen and mostly up to date. The conclusions are consistent with the presented evidence, the discussion refers to all important data reviewed in previous sections and is easy to follow. I would raise some points that could be better addressed by authors:

Response: Thank you for the positive evaluation and we have addressed the raised points as given below.

  1. Comment:   In lines 173-178 authors mentioned beneficial effects of pharmacists in ED regarding the cost reduction and increase of medicine appropriateness. Authors could refer to some specific findings in the cited literature as to provide some qualitative data concerning these improvements?

Response: As suggested, we have revised the manuscript to include specific findings.

“ considerable cost-avoidance at the Intermountain Medical Center, Utah.[66] ED pharmacists made 820 clinical interventions across 107 overnight shifts, leading to a substantial cost avoidance of $612,974, primarily due to direct patient care, ADE prevention and individualization of patient care” [Lines 177–179]

  1. Comment:  In lines 182-184 authors point out that some ED services are only available in larger, centrally located hospitals, as compared to smaller hospitals. Which services are less accessible and how it influences the quality of medicines use?  

Response: As suggested, we have incorporated the following points into the revised manuscript.

“In smaller hospitals, the availability of services such as round-the-clock pharmacist coverage and comprehensive MedRec is often limited due to resources and staffing constraints. These limitations can lead to delays in care, an increased risk of ADEs, and less comprehensive medication management.” [Lines 186–189]

  1. Comment: In lines 214-215 authors wrote that previous studies indicated that the use of automated dispensing systems may lead to occurrence of new types of medication errors? What kind of errors become more common upon introduction of these systems?

Response: As suggested, we have incorporated the following points into the revised manuscript.

“, including medication selection errors. Additionally, clinicians may bypass safety features or override alerts, causing override errors.” [Lines 227–229]

  1. Comment:In lines 318-325 authors described the Sticker model and its introduction. I think it could be interesting to the readers of Pharmacy to know how it worked out. What were the main findings of the cited study?

Response: As suggested, we have clarified this in the revised manuscript.

“This model, wherein pharmacists place stickers on medication charts to highlight potential issues or suggest therapeutic adjustments, was implemented to enhance communication between pharmacists and physicians. The study also demonstrated an improvement in the timeliness and accuracy of medication charts.” [Lines 342–345]

Reviewer 2 Report

Comments and Suggestions for Authors

While the article presents an interesting overview of the topic, I have some concerns regarding the methodology and structure of the review.

Strengths

The article is well-written and grammatically sound.

The content appears to be accurate and up-to-date.

The authors demonstrate a good understanding of the subject matter.

Major Concerns

1.           Review Methodology

The primary concern with this manuscript is that it follows a narrative review approach rather than a systematic review methodology. While narrative reviews can be valuable in certain contexts, they are generally considered less rigorous and more prone to bias than systematic reviews.

2.           Lack of Systematic Search Strategy

The authors do not present a clear, reproducible search strategy. A systematic review should include a detailed description of databases searched, search terms used, and inclusion/exclusion criteria for article selection.

3.           Potential for Selection Bias

Without a systematic approach, there is a risk of selection bias in the literature included. The authors may have inadvertently focused on studies that support their viewpoint while overlooking contradictory evidence.

4.           Absence of Quality Assessment

The manuscript lacks a formal quality assessment of the included studies. In a systematic review, each included study should be evaluated for methodological quality and risk of bias.

5.           Limited Synthesis of Evidence

The narrative approach provides a summary of existing literature but falls short in synthesizing the evidence in a systematic manner. A more structured approach to data extraction and synthesis would strengthen the review's conclusions.

Recommendations

1.   Consider revising the review methodology to follow a systematic approach, including ( you can opt for a scoping review):

             Developing a clear, reproducible search strategy

             Establishing explicit inclusion and exclusion criteria

             Conducting a formal quality assessment of included studies

             Providing a structured synthesis of the evidence

2.           If a narrative review is maintained, clearly state this in the title and methods section, and acknowledge the limitations of this approach in the discussion.

3.           Enhance the transparency of the review process by detailing how studies were selected and what criteria were used to evaluate their relevance and quality.

4.           Include a section on the limitations of the current review and suggestions for future research.

While the content of the review appears sound, the lack of a systematic approach significantly limits its scientific value and potential impact. I recommend major revisions to address these methodological issues before the manuscript can be considered for publication.

Author Response

While the article presents an interesting overview of the topic, I have some concerns regarding the methodology and structure of the review.

  1. Comment: Strengths
  • The article is well-written and grammatically sound.
  • The content appears to be accurate and up-to-date.
  • The authors demonstrate a good understanding of the subject matter.

Response: Thank you for the positive evaluation.

  1. Comment:

Major Concerns

Review Methodology

The primary concern with this manuscript is that it follows a narrative review approach rather than a systematic review methodology. While narrative reviews can be valuable in certain contexts, they are generally considered less rigorous and more prone to bias than systematic reviews.

  • Lack of Systematic Search Strategy - The authors do not present a clear, reproducible search strategy. A systematic review should include a detailed description of databases searched, search terms used, and inclusion/exclusion criteria for article selection.
  • Potential for Selection Bias - Without a systematic approach, there is a risk of selection bias in the literature included. The authors may have inadvertently focused on studies that support their viewpoint while overlooking contradictory evidence.
  • Absence of Quality Assessment - The manuscript lacks a formal quality assessment of the included studies. In a systematic review, each included study should be evaluated for methodological quality and risk of bias.
  • Limited Synthesis of Evidence - The narrative approach provides a summary of existing literature but falls short in synthesizing the evidence in a systematic manner. A more structured approach to data extraction and synthesis would strengthen the review's conclusions.

Recommendations

  1.  Consider revising the review methodology to follow a systematic approach, including ( you can opt for a scoping review):
  • Developing a clear, reproducible search strategy
  • Establishing explicit inclusion and exclusion criteria
  • Conducting a formal quality assessment of included studies
  • Providing a structured synthesis of the evidence
  1. If a narrative review is maintained, clearly state this in the title and methods section, and acknowledge the limitations of this approach in the discussion.
  2. Enhance the transparency of the review process by detailing how studies were selected and what criteria were used to evaluate their relevance and quality.
  3. Include a section on the limitations of the current review and suggestions for future research.

While the content of the review appears sound, the lack of a systematic approach significantly limits its scientific value and potential impact. I recommend major revisions to address these methodological issues before the manuscript can be considered for publication.

Response: Based on the comment, we have acknowledged the concerns raised regarding the narrative review approach and its potential limitations compared to a systematic review.

“4.3 Limitations

Our review is a narrative review in nature. Our focus was primarily on the literature pertinent to our central themes of QUMs in ED settings and on studies examining the services provided by ED pharmacists. We aimed was to highlight the most recent developments affecting medication safety and patient care in the ED setting. While narrative reviews provide summaries of existing literature and can be valuable in certain contexts, their lack of a reproducible search strategy renders them less rigorous and more prone to selection bias compared to systematic reviews. Future work should adopt a more structured method for data extraction and synthesis.” (Lines 501–508]

Reviewer 3 Report

Comments and Suggestions for Authors

To the authors

Nexus of Quality Use of Medicines, Pharmacists’ Activities and the Emergency Department: A Review

The review highlights pharmacists' role in the ED, particularly in medication reconciliation, error prevention, and improving patient care outcomes. The manuscript also discusses the challenges facing EDs, including overcrowding, increased demand, and the complexity of medication regimens, which remain in high interest for patients' safety and are a good paper for the scientific community with practical application.

However, some questions could be addressed to clarify or improve the manuscript.

1.    Inclusion criteria

o  How did you select the studies included in this narrative review? Were there specific criteria used for inclusion or exclusion of certain studies?

o  Why Pharmacy Technician-related studies were not included, any reason? Do you find the tasks related to a profession only or these activities can (must) be performed from a team perspective?

2.    Strategies for barriers

o  You mention barriers to the broader implementation of pharmacy services in the ED, such as staffing and resource constraints. Could you provide specific recommendations or strategies to overcome these challenges, especially in resource-limited settings?

3.    More information on studies

o  Can you provide more detailed data or case studies regarding the effectiveness of specific pharmacist-led medication charting models in real-world EDs?

4.    Future research

o  The manuscript discusses gaps in the existing literature, particularly around the MedRec (medication reconciliation) process. Could you elaborate on what specific future research or data would be needed to fill these gaps?

5.    Generalization possibilities

o  Given that many of the studies cited are from countries like Australia, the USA, and the UK, how applicable are the findings to other healthcare systems with different structures and resources?

6.    Training and certification

o  What additional training or certification would be necessary for pharmacists to take on expanded roles in the ED, such as medication charting or co-prescribing?

o  Only for pharmacists?

7.    How effective are the savings

o  The manuscript mentions cost savings associated with pharmacist interventions. Could you provide more detailed information on the methodology used to calculate these savings, and how they might vary across different hospital settings?

o  Only pharmacists interfere in the process of cost savings?

8.    Impact

o    Are there any long-term studies or follow-ups that examine patient outcomes beyond the immediate reduction in medication errors? Specifically, how do pharmacist interventions impact overall patient health and readmission rates?

Author Response

  1. Comment: The review highlights pharmacists' role in the ED, particularly in medication reconciliation, error prevention, and improving patient care outcomes. The manuscript also discusses the challenges facing EDs, including overcrowding, increased demand, and the complexity of medication regimens, which remain in high interest for patients' safety and are a good paper for the scientific community with practical application. However, some questions could be addressed to clarify or improve the manuscript.

Response: Thank you for the positive evaluation and we have addressed the concerns in the revised manuscript.

  1. Comment:Inclusion criteria – How did you select the studies included in this narrative review? Were there specific criteria used for inclusion or exclusion of certain studies?

Response: We did not set specific criteria for selecting studies because we chose to follow a narrative review instead of a systematic approach. We appreciate the chance to clear this up and have updated this in the revised version including the manuscript title.

“4.3 Limitations

Our review is a narrative review in nature. Our focus was primarily on literature pertinent to our central themes of QUMs in ED settings and on studies examining the services provided by ED pharmacists. Our aim was to highlight the most recent developments affecting medication safety and patient care in the ED setting. While narrative reviews provide summaries of existing literature and can be valuable in certain contexts, their lack of a reproducible search strategy renders them less rigorous and more prone to selection bias compared to systematic reviews. Future work should adopt a more structured method for data extraction and synthesis.” (Lines 501–508]

  1. Comment:Why Pharmacy Technician-related studies were not included, any reason? Do you find the tasks related to a profession only or these activities can (must) be performed from a team perspective?

Response: As noted in the title, our manuscript specifically focuses on services provided by pharmacists, either independently or in collaboration with physicians, as their clinical decision-making plays a vital role in enhancing medication safety in high-risk environments like emergency departments. In response to your suggestion, we have included the following points in our revised manuscript.

“Our review focuses on interventions where pharmacists contribute significantly to critical judgment and patient care outcomes. Although pharmacy technicians play an important role in medication preparation and distribution, optimal patient care often arises from team-based approaches. [20] Some of the pharmacists’ activities highlighted in our review could indeed be performed as part of an interdisciplinary team. Future reviews should investigate how the involvement of pharmacy technicians enhances ED pharmacy services, particularly regarding operational efficiency and support in ED settings.” {Lines 509–515]

  1. Comment: Strategies for barriers – You mention barriers to the broader implementation of pharmacy services in the ED, such as staffing and resource constraints. Could you provide specific recommendations or strategies to overcome these challenges, especially in resource-limited settings?

Response: Based on the comment, we have added the following ones in our revision.

“Staffing and resource constraints present significant barriers to the broader implementation of ED pharmacy services, particularly in resource-limited settings. The services could be optimized by reallocating specific tasks, for example, distributing non-clinical tasks to pharmacy technicians and higher-level clinical interventions to pharmacists. Tele pharmacy services could also be leveraged to extend the reach of pharmacists into EDs in resource limited areas.” [Lines 195–199]

  1. Comment:More information on studies – Can you provide more detailed data or case studies regarding the effectiveness of specific pharmacist-led medication charting models in real-world EDs?

Response: Detailed information about the effectiveness of a medication charting model can be found on page 10 (lines 388–410) and in Figure 4 of the original manuscript. We believe this provides sufficient detail. However, in line with the reviewer's suggestion, we have added the phrase “… conducted in a real-world ED setting” in the revised version (line 389).

  1. Comment:Future research – The manuscript discusses gaps in the existing literature, particularly around the MedRec (medication reconciliation) process. Could you elaborate on what specific future research or data would be needed to fill these gaps?

Response: Based on the comment, we have included the following in our revision:

“The existing literature lacks consensus on best MedRec practices, and variability in MedRec implementation can impact its effectiveness. Future research should focus on developing and evaluating standardized MedRec protocols specifically tailored for ED settings.” [Lines 322–325]

  1. Comment:Generalization possibilities – Given that many of the studies cited are from countries like Australia, the USA, and the UK, how applicable are the findings to other healthcare systems with different structures and resources?

Response: As suggested, we have revised our manuscript to reflect these considerations more clearly, as shown below.

“While the core principles of ED-based pharmacy services, such as improving medication safety and reducing errors, are broadly applicable across countries and settings, we acknowledge that the implementation and effectiveness of these services may vary across different healthcare environments. Factors such as staffing levels, access to technology, regulatory frameworks and interprofessional collaboration models can influence how feasible and impactful these services are in other countries. Future research and localized studies are essential to ensure the effective adaption of pharmacist interventions in other ED settings.” [Lines 516–523]

  1. Comment:Training and certification – What additional training or certification would be necessary for pharmacists to take on expanded roles in the ED, such as medication charting or co-prescribing? Only for pharmacists?

Response: We appreciate your suggestion, and we have added the following points into the revised manuscript.

“Pharmacists need additional training and certification to assume expanded roles in medication charting and co-prescribing in E.Ds. Typically, this may involve advanced clinical training, such as a medication charting credentialing program. [128] Moreover, collaborative practice agreements or co-prescribing privileges may necessitate certification in drug therapy management or completion of further training in patient assessment and pharmacotherapy.” (Lines 377–382]

  1. Comment:How effective are the savings – The manuscript mentions cost savings associated with pharmacist interventions. Could you provide more detailed information on the methodology used to calculate these savings, and how they might vary across different hospital settings? Only pharmacists interfere in the process of cost savings?

Response: As suggested, in the revised manuscript, we have elaborated on how cost savings were typically estimated in the studies reviewed.

“Cost savings estimations often include calculating the cost avoidance associated with preventing medication errors, or reducing adverse drug events. Cost savings were generally based on the estimated costs of managing preventable errors or adverse outcomes, such as prolonged hospital stays.[131] These savings can vary significantly across different hospital settings, depending on factors such as patient demographics, hospital size, available resources, or the specific scope of pharmacist involvement.” {Lines 404–410]

  1. Comment:Impact – Are there any long-term studies or follow-ups that examine patient outcomes beyond the immediate reduction in medication errors? Specifically, how do pharmacist interventions impact overall patient health and readmission rates?

Response: As suggested, we have revised the manuscript to include examples of these long-term studies where available.

“Most studies emphasize short-term outcomes, such as the immediate reduction of medication errors. However, there is a need for data from longitudinal studies and follow-ups that assess the long-term effects on patient health, hospital readmission rates, and overall healthcare utilization. Future investigations should aim to provide more robust evidence on how ED pharmacist services contribute to improving patient outcomes over extended periods.” [Lines 524–529]

Round 2

Reviewer 2 Report

Comments and Suggestions for Authors

Dear Authors , despite your changes, it is my understanding that a narrative review is not the appropriate approach. The contribution of a narrative review to the field may also be limited. Given the abundance of published literature, such a review may not offer substantial new insights or significantly advance the discipline. The scientific community and journals are increasingly prioritizing publications that provide novel data or analyses. Lastly, complex topics may be oversimplified in narrative reviews, potentially leading to misinterpretation or overgeneralization of research findings.